# Essential Oils in Citrus Fruit Ripening and Postharvest Quality

Maria Michela Salvatore [1,2] , Rosario Nicoletti [3,4,*] and Anna Andolfi [1,5]

1    Department of Chemical Sciences, University of Naples Federico II, 80126 Naples, Italy;
     mariamichela.salvatore@unina.it (M.M.S.); andolfi@unina.it (A.A.)
2    Institute for Sustainable Plant Protection, National Research Council, 80055 Portici, Italy
3    Department of Agricultural Sciences, University of Naples Federico II, 80055 Portici, Italy
4    Council for Agricultural Research and Economics, Research Center for Olive, Fruit and Citrus Crops,
     81100 Caserta, Italy
5    BAT Center—Interuniversity Center for Studies on Bioinspired Agro-Environmental Technology,
     University of Naples Federico II, 80055 Portici, Italy
*    Correspondence: rosario.nicoletti@crea.gov.it

**Abstract:** Citrus essential oils (EOs) are widely used as flavoring agents in food, pharmaceutical, cosmetical and chemical industries. For this reason, their demand is constantly increasing all over the world. Besides industrial applications, the abundance of EOs in the epicarp is particularly relevant for the quality of citrus fruit. In fact, these compounds represent a natural protection against postharvest deteriorations due to their remarkable antimicrobial, insecticidal and antioxidant activities. Several factors, including genotype, climatic conditions and cultural practices, can influence the assortment and accumulation of EOs in citrus peels. This review is focused on factors influencing variation of the EOs' composition during ripening and on the implications on postharvest quality of the fruit.

**Keywords:** citrus peel; terpenes; maturation stages; antimicrobial effects; insecticidal effects; fruit quality

## 1. Introduction

The genus *Citrus* (Rutaceae, Aurantioideae) includes four basic taxa and several hybrid species which are mainly cultivated in subtropical regions [1]. All *Citrus* species are characterized by a particular kind of berry fruit, the hesperidium, of which the epicarp is scattered with cavities lined with secretory cells producing essential oils (EOs). Besides conferring the typical scent, these mixtures possess a series of notable biological properties which contribute to the fruit quality and represent an added value for these crops in view of several possible applications in the food, pharmaceutical, cosmetical and chemical industries [2].

One of the most important applications is based on the antimicrobial properties of EOs, which have been recognized for a long time and recently boosted by the urgent need for alternatives to chemical bactericides following the spread of antibiotic resistance. Generally recognized as safe, these compounds possess inhibitory properties against a wide range of microorganisms, both as direct oil and in vapor form; undoubtedly, they represent a group of natural antimicrobials which may fulfil health and technical requirements for both the consumers and the food industry [3].

## 2. Essential Oils of Citrus Fruits

According to the International Organization for Standardization (ISO), "essential oil" is defined as a "product obtained from a natural raw material of plant origin, by steam distillation, by mechanical processes from the epicarp of citrus fruit or by dry distillation, after separation of the aqueous phase—if any—by physical processes" [4].

Essential oils are composed of lipophilic and highly volatile secondary metabolites, with a molecular weight below 300 Da, that can be physically separated from other plant components [5].

In citrus, essential oils could be extracted from leaf, flower, epicarp and fruit juice representing more than 80% of the volatile fraction, mainly constituted of terpenoids, phenylpropanoids and short-chain aliphatic hydrocarbon derivatives [6]. Terpenoids are the predominant EO constituents; they are characterized by a wide structural diversity deriving from $C_5$ isoprene units and are classified as hemiterpenes ($C_5$), monoterpenes ($C_{10}$), sesquiterpenes ($C_{15}$), diterpenes ($C_{20}$), sesterterpenes ($C_{25}$), triterpenes ($C_{30}$) and tetraterpenes ($C_{40}$) [7]. The monoterpene hydrocarbons and oxygenated monoterpenes comprising alcohols, aldehydes, ketones and esters, are responsible for the odor and flavor profiles of fruit. Despite the dominance of the monoterpene hydrocarbon limonene in the citrus EO composition, other less abundant monoterpenes actively contribute to the citrus aroma, representing major determinants of fruit quality.

EO composition is known to vary among the several *Citrus* species, according to both the genetic bases and a series of environmental factors which are examined in the next section. Compounds characterizing EOs are listed in Table 1, based on data gathered from recent literature concerning the most common citrus species, such as: key lime (*C. aurantifolia*) [8,9], bitter orange (*C. aurantium*) [10–12], bergamot (*C. bergamia*) [13–15], pomelo (*C. maxima* or *C. grandis*) [16,17], kaffir lime (*C. hystrix*) [18], persian lime (*C. latifolia*) [8], sweet lemon (*C. limetta*) [19,20], lemon (*C. limon*) [8,11,21], rangpur (*C. limonia*) [8], wild orange (*C. macroptera*) [16], citron (*C. medica*) [22,23], grapefruit (*C. paradisi*) [17], mandarin orange (*C. reticulata*) [11,24–26], sweet orange (*C. sinensis*) [11,27], hyuganatsu (*C. tamurana*) [28,29] and satsuma mandarin (*C. unshiu*) [30].

**Table 1.** Essential oils reported from peels of *Citrus* fruits.

| Compound * | *Citrus* Species | Structure |
|---|---|---|
| Monoterpenes | | |
| Alcohols and derivatives | | |
| Borneol | *C. aurantium* [11], *C. limon* [11,21], *C. reticulata* [11], *C. sinensis* [11], *C. tamurana* [28] |  |
| Campherenol | *C. bergamia* [14] |  |
| Carvacrol | *C. aurantium* [11], *C. limon* [11], *C. reticulata* [11], *C. sinensis* [11], *C. tamurana* [28] |  |
| Carveol | *C. limetta* [19], *C. medica* [22] |  |
| *trans*-Carveol | *C. limon* [20,21], *C. maxima* [16], *C. paradisi* [17], *C. tamurana* [29] |  |

**Table 1.** *Cont.*

| Compound * | *Citrus* Species | Structure |
|---|---|---|
| *cis*-Carveol | *C. limetta* [20], *C. maxima* [16,17], *C. paradisi* [17], *C. tamurana* [28] | |
| Citronellol (= citronellyl alcohol) | *C. aurantium* [11], *C. bergamia* [14],*C. limon* [11], *C. macroptera* [16], *C. medica* [22,23], *C. reticulata* [11,24,26], *C. sinensis* [11], *C. tamurana* [28,29] | |
| α-Citronellol | *C. reticulata* [26] | |
| D-Citronellol | *C. hystrix* [18] | |
| Dehydrocarveol | *C. maxima* [17], *C. paradisi* [17], *C. tamurana* [28] | |
| Geraniol (=geranyl alcohol) | *C. aurantifolia* [8], *C. latifolia* [8], *C. limetta* [21], *C. limon* [8,11], *C. macroptera* [16], *C. maxima* [17], *C. medica* [23], *C. paradisi* [17], *C. reticulata* [11,25], *C. tamurana* [28,29] | |
| Geraniol methyl ether | *C. medica* [22] | |
| Nerol (=Z-geraniol) | *C. aurantium* [10–12], *C. aurantifolia* [8], *C. bergamia* [14,15], *C. latifolia* [8],*C. limetta* [19], *C. limon* [8,11], *C. maxima* [17], *C. medica* [23], *C. paradisi* [17], *C. reticulata* [11,25,26], *C. sinensis* [11], *C. tamurana* [28,29] | |
| Isopinocarveol (=pinocarveol) | *C. limetta* [19] | |
| Limonene-1,2-diol | *C. tamurana* [28,29], *C. medica* [22,23] | |
| Linalool | *C. aurantifolia* [8], *C. aurantium* [10–12], *C. bergamia* [13–15], *C. hystrix* [18], *C. latifolia* [8], *C. limetta* [19,20], *C. limon* [8,11,21], *C. limonia* [8], *C. macroptera* [16], *C. maxima* [16,17], *C. medica* [22,23], *C. paradisi* [17], *C. reticulata* [11,24,25], *C. sinensis* [11,27], *C. tamurana* [28,29], *C. unshiu* [30] | |
| L-Menthol | *C. tamurana* [28,29] | |

**Table 1.** *Cont.*

| Compound * | *Citrus* Species | Structure |
|---|---|---|
| *trans-p*-Mentha-2,8-dienol | *C. maxima* [17], *C. paradisi* [17] | |
| *trans-p*-Menth-2,8-dien-1-ol | *C. limetta* [20], *C. maxima* [16,17], *C. medica* [23], *C. paradisi* [17], *C. tamurana* [28] | |
| *cis-p*-Menth-2,8-dien-1-ol | *C. limetta* [20] | |
| *p*-Menth-2,8-dien-1-ol | *C. limon* [21] | |
| *p*-Mentha-1,8-dien-10-ol | *C. tamurana* [28,29] | |
| *p*-Mentha-1-en-9-ol | *C. maxima* [17], *C. paradisi* [17], *C. tamurana* [28,29] | |
| Myrcenol | *C. aurantium* [10], *C. tamurana* [28] | |
| *trans*-Myrtanol | *C. limetta* [19] | |
| 3,7-Nonadien-2-ol, 4,8-dimethyl | *C. medica* [22] | |
| Perillol | *C. limon* [21], *C. maxima* [17], *C. paradisi* [17] | |

**Table 1.** *Cont.*

| Compound * | *Citrus* Species | Structure |
|---|---|---|
| *trans*-Piperitol | *C. tamurana* [28] | |
| *cis*-Piperitol | *C. maxima* [17], *C. paradisi* [17] | |
| Sabinol | *C. limon* [21] | |
| *trans*-Sabinene hydrate | *C. bergamia* [15], *C. limetta* [19], *C. paradisi* [17], *C. sinensis* [27] | |
| *cis*-Sabinene hydrate | *C. aurantium* [11,12], *C. bergamia* [14], *C. limetta* [19], *C. limon* [11], *C. reticulata* [11], *C. sinensis* [11] | |
| Terpinen-4-ol | *C. limetta* [19], *C. aurantifolia* [8], *C. limonia* [8], *C. latifolia* [8], *C. macroptera* [16], *C. maxima* [16], *C. hystrix* [18], *C. limon* [8,11,21], *C. reticulata* [11,24], *C. aurantium* [11,12], *C. sinensis* [11,27], *C. tamurana* [28,29], *C. bergamia* [14,15], *C. medica* [22,23] | |
| α-Terpineol | *C. aurantifolia* [8,9], *C. aurantium* [10–12], *C. bergamia* [14,15], *C. hystrix* [18], *C. latifolia* [8], *C. limetta* [19], *C. limon* [8,11,21], *C. limonia* [8], *C. macroptera* [16], *C. medica* [22], *C. paradisi* [17], *C. reticulata* [11,24,25], *C. sinensis* [11,27], *C. tamurana* [28,29] | |
| Thymol | *C. reticulata* [26] | |
| *cis*-Verbenol | *C. medica* [23] | |
| *trans*-Verbenol | *C. limon* [21] | |

**Table 1.** *Cont.*

| Compound * | *Citrus* Species | Structure |
|---|---|---|
| Aldehydes | | |
| Citronellal | *C. aurantium* [12], *C. bergamia* [14], *C. hystrix* [18], *C. limon* [21], *C. limonia* [8], *C. maxima* [17], *C. paradisi* [17], *C. reticulata* [24,25], *C. sinensis* [27], *C. tamurana* [28,29] | |
| Cumin aldehyde | *C. maxima* [17], *C. paradisi* [17], *C. tamurana* [28] | |
| Geranial (=*E*-citral) | *C. aurantifolia* [8], *C. bergamia* [14,15], *C. latifolia* [8], *C. limetta* [19], *C. limon* [8], *C. macroptera* [16], *C. maxima* [16,17], *C. medica* [22,23], *C. paradisi* [17], *C. reticulata* [24,25], *C. sinensis* [27] | |
| Neral (=*Z*-citral) | *C. aurantifolia* [8], *C. aurantium* [12], *C. bergamia* [14,15], *C. latifolia* [8], *C. limetta* [19], *C. limon* [11,21], *C. macroptera* [16], *C. maxima* [17], *C. medica* [22], *C. paradisi* [17], *C. reticulata* [25], *C. sinensis* [27], *C. tamurana* [28,29] | |
| Perillaldehyde | *C. bergamia* [14], *C. limetta* [19], *C. maxima* [17], *C. paradisi* [17], *C. reticulata* [26], *C. reticulata* [24], *C. tamurana* [28,29] | |
| Esters | | |
| Bornyl acetate | *C. aurantium* [11], *C. limon* [11], *C. reticulata* [11], *C. sinensis* [11], *C. tamurana* [28,29] | |
| Carveol propionate | *C. limon* [21] | |
| Citronellol acetate | *C. bergamia* [14,15], *C. hystrix* [18], *C. reticulata* [25], *C. tamurana* [28] | |
| Citronellol formate | *C. macroptera* [16], *C. tamurana* [28,29] | |
| Geraniol acetate | *C. aurantium* [11,12], *C. bergamia* [14,15], *C. latifolia* [8], *C. limetta* [19], *C. limon* [8,11], *C. medica* [22], *C. reticulata* [11,25], *C. sinensis* [11], *C. tamurana* [28,29] | |
| Geraniol formate | *C. reticulata* [25] | |

**Table 1.** *Cont.*

| Compound * | *Citrus* Species | Structure |
|---|---|---|
| Geraniol propionate | *C. maxima* [17], *C. paradisi* [17], *C. tamurana* [28,29] | |
| Isobornyl acetate | *C. bergamia* [14] | |
| Linalool acetate (=bergamiol) | *C. aurantium* [11,12], *C. bergamia* [13–15], *C. limetta* [19], *C. limon* [11], *C. maxima* [17], *C. paradisi* [17], *C. reticulata* [11], *C. sinensis* [11], *C. tamurana* [28,29] | |
| Linalool butyrate | *C. aurantium* [10] | |
| *p*-Mentha-1-en-9-yl acetate | *C. tamurana* [28] | |
| Methyl geranate | *C. bergamia* [14] | |
| Methylthymol | *C. reticulata* [24] | |
| *Z*-2-Octen-1-ol,3,7-dimethyl-, isobutyrate | *C. medica* [22] | |
| Perillyl acetate | *C. bergamia* [15], *C. paradisi* [17] | |
| Terpineol acetate (=terpinyl acetate) | *C. aurantium* [11,12], *C. bergamia* [14,15], *C. limetta* [19], *C. limon* [11], *C. medica* [23], *C. reticulata* [11], *C. sinensis* [11], *C. tamurana* [28,29] | |

**Table 1.** *Cont.*

| Compound * | *Citrus* Species | Structure |
|---|---|---|
| β-Terpinyl acetate | *C. medica* [23] | |
| Hydrocarbons | | |
| Camphene | *C. aurantium* [11,12], *C. bergamia* [14,15], *C. hystrix* [18], *C. latifolia* [8], *C. limetta* [19,20], *C. limon* [8,11], *C. reticulata* [11], *C. sinensis* [11], *C. tamurana* [28,29] | |
| 2-Carene | *C. reticulata* [24] | |
| 3-Carene | *C. aurantium* [10,11], *C. bergamia* [14], *C. limon* [11,21], *C. medica* [22], *C. reticulata* [11], *C. sinensis* [11,27], *C. tamurana* [28,29] | |
| *o*-Cymene | *C. aurantium* [12], *C. limon* [21], *C. unshiu* [30] | |
| *m*-Cymene | *C. latifolia* [8], *C. limonia* [8] | |
| *p*-Cymene | *C. aurantifolia* [8], *C. aurantium* [11,12], *C. bergamia* [14,15], *C. latifolia* [8], *C. limetta* [20], *C. limon* [8,11], *C. limonia* [8], *C. reticulata* [11], *C. sinensis* [11,27], *C. tamurana* [28,29] | |
| α-Fenchene | *C. aurantium* [12], *C. limon* [21], *C. tamurana* [28,29] | |
| Isolimonene | *C. medica* [23] | |
| Limonene | *C. medica* [23], *C. latifolia* [8], *C. macroptera* [16], *C. paradisi* [17], *C. limon* [8,11,21], *C. aurantium* [10–12], *C. sinensis* [8,11,27], *C. bergamia* [13–15], *C. reticulata* [11,25], *C. tamurana* [28,29], *C. maxima* [16,17], *C. aurantifolia* [8,9] | |
| D-Limonene | *C. hystrix* [18], *C. limetta* [19,20], *C. medica* [22,23], *C. reticulata* [24] | |

**Table 1.** *Cont.*

| Compound * | *Citrus* Species | Structure |
|---|---|---|
| L-Limonene | *C. unshiu* [30] | |
| Myrcene (= *β*-myrcene) | *C. aurantifolia* [8], *C. aurantium* [10–12], *C. bergamia* [14,15], *C. latifolia* [8], *C. limetta* [19,20], *C. limon* [11], *C. limonia* [8], *C. macroptera* [16], *C. maxima* [16], *C. medica* [22,23], *C. reticulata* [11,24,25], *C. sinensis* [8,11,27], *C. tamurana* [28,29], *C. unshiu* [30] | |
| Allo-Ocimene | *C. medica* [22] | |
| *E*-*β*-Ocimene | *C. aurantium* [10,11], *C. bergamia* [14], *C. limetta* [11], *C. reticulata* [11,24,25], *C. sinensis* [11,27] | |
| *Z*-*β*-Ocimene | *C. bergamia* [14,15], *C. medica* [22], *C. reticulata* [25], *C. tamurana* [28] | |
| Z-1,3,6-Octatriene,3,7-dimethyl- | *C. medica* [23] | |
| *α*-Phellandrene | *C. medica* [22], *C. latifolia* [8], *C. bergamia* [14,15], *C. tamurana* [28,29] | |
| *β*-Phellandrene | *C. bergamia* [15], *C. reticulata* [24], *C. sinensis* [27], *C. tamurana* [28] | |
| *α*-Pinene | *C. aurantifolia* [8,9], *C. aurantium* [10–12], *C. bergamia* [14,15], *C. hystrix* [18], *C. latifolia* [8], *C. limetta* [19,20], *C. limon* [8,11], *C. limonia* [8], *C. macroptera* [16], *C. maxima* [16,17], *C. medica* [22,23], *C. paradisi* [17], *C. reticulata* [11,24,25], *C. sinensis* [11,27], *C. tamurana* [28,29] | |
| *β*-Pinene | *C. limetta* [19], *C. medica* [22], *C. limonia* [8], *C. latifolia* [8], *C. macroptera* [16], *C. hystrix* [18], *C. paradisi* [17], *C. aurantium* [10–12], *C. bergamia* [13–15], *C. limon* [11,21], *C. reticulata* [11,24], *C. sinensis* [11,27], *C. aurantifolia* [8,9], *C. tamurana* [28,29], *C. maxima* [16,17] | |
| Sabinene | *C. aurantifolia* [8,9], *C. aurantium* [11,12], *C. bergamia* [14,15], *C. limetta* [19], *C. limon* [8,11], *C. limonia* [8], *C. macroptera* [16], *C. maxima* [16,17], *C. paradisi* [17], *C. reticulata* [11,25], *C. sinensis* [11,27] | |
| D-Sabinene | *C. tamurana* [28,29] | |

**Table 1.** *Cont.*

| Compound * | *Citrus* Species | Structure |
|---|---|---|
| α-Terpinene | *C. aurantifolia* [8,9], *C. aurantium* [11], *C. bergamia* [14,15], *C. latifolia* [8], *C. limon* [8,11], *C. limonia* [8], *C. maxima* [17], *C. medica* [22], *C. paradisi* [17], *C. reticulata* [11], *C. sinensis* [11], *C. tamurana* [28,29] | |
| γ-Terpinene | *C. aurantifolia* [8,9], *C. aurantium* [11,12], *C. bergamia* [13–15], *C. hystrix* [18], *C. latifolia* [8], *C. limon* [8,11,21], *C. limonia* [8], *C. macroptera* [16], *C. maxima* [16,17], *C. medica* [22], *C. paradisi* [17], *C. reticulata* [11,24], *C. sinensis* [11,27], *C. tamurana* [28,29], *C. unshiu* [30] | |
| Terpinolene | *C. aurantifolia* [8], *C. aurantium* [11], *C. bergamia* [14,15], *C. latifolia* [8], *C. limon* [11], *C. limonia* [8], *C. paradisi* [17], *C. reticulata* [11,24], *C. sinensis* [8,11,27], *C. tamurana* [28,29] | |
| α-Thujene | *C. aurantifolia* [8], *C. aurantium* [11,12], *C. bergamia* [14], *C. hystrix* [18], *C. latifolia* [8], *C. limon* [8,11], *C. limonia* [8], *C. paradisi* [17], *C. reticulata* [11,24], *C. sinensis* [11,27], *C. tamurana* [28,29] | |
| β-Thujene | *C. tamurana* [28,29] | |
| Tricyclene | *C. aurantium* [11,12], *C. bergamia* [14], *C. limon* [11], *C. reticulata* [11], *C. sinensis* [11] | |
| Ketones | | |
| Camphor | *C. aurantium* [11], *C. limetta* [19], *C. limon* [11,21], *C. reticulata* [11,24], *C. sinensis* [11] | |
| δ-Camphor | *C. tamurana* [28,29] | |
| Carvomenthone | *C. medica* [22] | |
| ʟ-Carvone | *C. tamurana* [28,29] | |

**Table 1.** *Cont.*

| Compound * | *Citrus* Species | Structure |
|---|---|---|
| Carvone | *C. bergamia* [15], *C. maxima* [16,17], *C. paradisi* [17], *C. reticulata* [25] | |
| *cis*-Dihydrocarvone | *C. aurantium* [11], *C. limon* [11], *C. reticulata* [11], *C. sinensis* [11] | |
| Isopiperitone | *C. tamurana* [28,29] | |
| Menthone | *C. tamurana* [28,29] | |
| Sabina ketone | *C. maxima* [17], *C. paradisi* [17] | |
| 3-Terpinolenone (=piperitone) | *C. limon* [21] | |
| Oxides | | |
| 4,5-Epoxycarene | *C. medica* [23] | |
| Carvone oxide | *C. tamurana* [28] | |
| 1,8-Cineole (= eucalyptol) | *C. aurantifolia* [9], *C. aurantium* [11,12], *C. bergamia* [15], *C. limon* [11], *C. macroptera* [16], *C. reticulata* [11,25], *C. sinensis* [11] | |

**Table 1.** *Cont.*

| Compound * | *Citrus* Species | Structure |
|---|---|---|
| *cis*-Limonene-1,2-epoxide | *C. bergamia* [14], *C. maxima* [16], *C. reticulata* [25], *C. sinensis* [27], *C. tamurana* [28] | |
| *trans*-Limonene-1,2,-epoxide | *C. bergamia* [14,15], *C. limon* [21], *C. reticulata* [25], *C. sinensis* [27], *C. tamurana* [28] | |
| Z-Linalool pyranoxide | *C. tamurana* [28,29] | |
| *cis*-Linalool-oxide | *C. aurantium* [10,11], *C. bergamia* [15], *C. hystrix* [18], *C. limetta* [20], *C. limon* [11], *C. macroptera* [16], *C. maxima* [16], *C. reticulata* [11], *C. sinensis* [11], *C. tamurana* [28,29] | |
| *trans*-Linalool oxide | *C. aurantium* [12], *C. bergamia* [15], *C. hystrix* [18], *C. macroptera* [16], *C. maxima* [16], *C. tamurana* [28,29] | |
| Myrcene epoxide | *C. paradisi* [17] | |
| Nerol oxide | *C. tamurana* [28,29] | |
| 7-Oxabicycloheptane, 1-methyl-4-(1-methylethyl) | *C. medica* [23] | |
| Perillene | *C. paradisi* [17] | |
| α-Pinene oxide | *C. limon* [21] | |
| Rose furan epoxide | *C. reticulata* [25] | |

**Table 1.** *Cont.*

| Compound * | *Citrus* Species | Structure |
|---|---|---|
| | Sesquiterpenes | |
| | Alcohols and derivatives | |
| α-Bisabolol | *C. bergamia* [14,15], *C. latifolia* [8], *C. limetta* [19], *C. limon* [21], *C. medica* [22,23], *C. tamurana* [28,29] | |
| β-Bisabolol | *C. limon* [21], *C. limetta* [19], *C. medica* [22,23] | |
| α-Cadinol | *C. macroptera* [16], *C. tamurana* [28] | |
| Cedrenol | *C. tamurana* [28] | |
| Cedrol | *C. bergamia* [15], *C. maxima* [17], *C. paradisi* [17], *C. tamurana* [28,29] | |
| Cubenol | *C. macroptera* [16] | |
| Elemol | *C. aurantifolia* [9], *C. hystrix* [18], *C. macroptera* [16], *C. maxima* [17], *C. paradisi* [17], *C. tamurana* [28,29] | |
| α-Eudesmol | *C. aurantifolia* [9] | |
| β-Eudesmol | *C. aurantifolia* [9], *C. macroptera* [16], *C. tamurana* [28,29] | |

| Compound * | *Citrus* Species | Structure |
|---|---|---|
| γ-Eudesmol | *C. aurantifolia* [9], *C. tamurana* [28] | |
| 7-epi-α-Eudesmol | *C. aurantifolia* [9] | |
| 2*E*,6*E*-Farnesol | *C. limetta* [19], *C. tamurana* [28,29] | |
| 2*Z*,6*E*-Farnesol | *C. aurantium* [11], *C. limon* [11], *C. paradisi* [17], *C. reticulata* [11], *C. sinensis* [11], *C. tamurana* [28,29] | |
| Globulol | *C. paradisi* [17], *C. tamurana* [28,29] | |
| Ledol | *C. limon* [21] | |
| *E*-Nerolidol | *C. reticulata* [25], *C. aurantium* [12], *C. bergamia* [15], *C. macroptera* [16], *C. tamurana* [28,29] | |
| *Z*-Nerolidol | *C. paradisi* [17], *C. reticulata* [26], *C. tamurana* [28,29] | |
| *α*-Santalol | *C. limetta* [19] | |
| *E*-*β*-santalol | *C. limetta* [19] | |
| *E*-sesquisabinene hydrate | *C. bergamia* [14] | |

**Table 1.** *Cont.*

| Compound * | *Citrus* Species | Structure |
|---|---|---|
| Spathulenol | *C. aurantium* [11], *C. limon* [11], *C. reticulata* [11], *C. sinensis* [11], *C. tamurana* [28,29] | |
| Valerianol | *C. aurantifolia* [9] | |
| Viridiflorol | *C. tamurana* [28] | |
| Aldehydes | | |
| α-Sinensal | *C. maxima* [17], *C. paradisi* [17], *C. reticulata* [26] | |
| β-Sinensal | *C. maxima* [17], *C. paradisi* [17], *C. reticulata* [26], *C. tamurana* [28] | |
| Hydrocarbons | | |
| Aromadendrene | *C. limetta* [19] | |
| α-Bergamotene | *C. bergamia* [15] | |
| *cis*-α-Bergamotene | *C. latifolia* [8], *C. limon* [21] | |
| *trans*-α-Bergamotene | *C. aurantifolia* [8], *C. bergamia* [14], *C. latifolia* [8], *C. limetta* [19], *C. limon* [8], *C. limonia* [8], *C. medica* [22,23], *C. reticulata* [25] | |
| Bicyclogermacrene | *C. bergamia* [14], *C. macroptera* [16] | |

**Table 1.** *Cont.*

| Compound * | *Citrus* Species | Structure |
|---|---|---|
| α-Bisabolene | *C. latifolia* [8], *C. limetta* [21] | |
| β-Bisabolene | *C. aurantifolia* [8], *C. bergamia* [14,15], *C. latifolia* [8], *C. limon* [8], *C. limonia* [8], *C. reticulata* [25] | |
| E-γ-Bisabolene | *C. bergamia* [14] | |
| Z-γ-Bisabolene | *C. bergamia* [14] | |
| β-Cadinene | *C. hystrix* [18] | |
| δ-Cadinene | *C. aurantifolia* [8], *C. aurantium* [12], *C. limonia* [8], *C. macroptera* [16], *C. medica* [22,23], *C. tamurana* [28] | |
| α-Cedrene | *C. maxima* [17], *C. paradisi* [17], *C. tamurana* [28,29] | |
| α-Caryophyllene (=humulene) | *C. aurantifolia* [8], *C. aurantium* [11,12], *C. bergamia* [14], *C. hystrix* [18], *C. latifolia* [8], *C. limon* [11], *C. macroptera* [16], *C. medica* [23], *C. reticulata* [11,25], *C. sinensis* [11], *C. tamurana* [28,29] | |
| β-Caryophyllene | *C. aurantifolia* [8], *C. aurantium* [12], *C. bergamia* [14,15], *C. hystrix* [18], *C. latifolia* [8], *C. limetta* [8], *C. limon* [8], *C. macroptera* [16], *C. maxima* [17], *C. medica* [22,23], *C. paradisi* [17], *C. reticulata* [25], *C. tamurana* [28,29] | |
| α-Copaene | *C. aurantium* [12], *C. hystrix* [18], *C. macroptera* [16], *C. maxima* [17], *C. paradisi* [17], *C. tamurana* [28,29] | |

**Table 1.** *Cont.*

| Compound * | *Citrus* Species | Structure |
|---|---|---|
| α-Cubebene | *C. maxima* [17], *C. paradisi* [17], *C. tamurana* [28] | |
| β-Cubebene | *C. hystrix* [18], *C. macroptera* [16], *C. tamurana* [28,29] | |
| α-Curcumene | *C. limon* [21] | |
| β-Curcumene | *C. limon* [21] | |
| β-Elemene | *C. aurantifolia* [8], *C. aurantium* [12], *C. hystrix* [18], *C. latifolia* [8], *C. macroptera* [16], *C. sinensis* [27], *C. tamurana* [28,29], *C. unshiu* [30] | |
| δ-Elemene | *C. aurantifolia* [8], *C. aurantium* [12], *C. bergamia* [14], *C. maxima* [17], *C. paradisi* [17], *C. reticulata* [24], *C. reticulata* [25] | |
| γ-Elemene | *C. reticulata* [24], *C. aurantium* [12], *C. tamurana* [28,29] | |
| E,E-α-Farnesene | *C. limetta* [19], *C. macroptera* [16], *C. maxima* [17], *C. medica* [23], *C. paradisi* [17], *C. unshiu* [30] | |
| E-β-Farnesene | *C. aurantifolia* [8], *C. bergamia* [15], *C. latifolia* [8], *C. limetta* [19], *C. tamurana* [28,29] | |
| Z-β-Farnesene | *C. bergamia* [14], *C. medica* [23], *C. tamurana* [28] | |
| Germacrene B | *C. reticulata* [25,26] | |

**Table 1.** *Cont.*

| Compound * | *Citrus* Species | Structure |
|---|---|---|
| Germacrene D | *C. aurantifolia* [8], *C. aurantifolia* [8], *C. aurantium* [11], *C. bergamia* [14,15], *C. hystrix* [18], *C. latifolia* [8], *C. limon* [8,11,21], *C. macroptera* [16], *C. medica* [22,23], *C. reticulata* [11,24], *C. sinensis* [11], *C. tamurana* [28,29] | |
| α-Muurolene (=α-Cadinene) | *C. macroptera* [16], *C. medica* [22] | |
| Z-β-Santalene | *C. bergamia* [14], *C. limetta* [19], *C. limon* [21] | |
| epi-β-Santalene | *C. limetta* [19] | |
| Sesquiphellandrene | *C. bergamia* [14], *C. limetta* [21], *C. tamurana* [28,29] | |
| Sesquithujene | *C. bergamia* [14] | |
| Seychellene | *C. limon* [21] | |
| Valecene | *C. aurantium* [11], *C. limon* [11], *C. reticulata* [11], *C. sinensis* [11], *C. tamurana* [28,29] | |
| α-Ylangene | *C. tamurana* [28] | |
| β-Ylangene | *C. tamurana* [29] | |
| Zizaene | *C. limon* [21] | |

**Table 1.** *Cont.*

| Compound * | *Citrus* Species | Structure |
|---|---|---|
| Esters | | |
| Cedryl acetate | *C. tamurana* [28] |  |
| 2E,6E-Farnesol acetate | *C. aurantium* [12], *C. tamurana* [28,29] |  |
| Methoprene | *C. medica* [22] |  |
| Nerolidol acetate | *C. maxima* [17], *C. paradisi* [17] |  |
| Nerol acetate | *C. aurantifolia* [8], *C. aurantium* [12], *C. bergamia* [14,15], *C. latifolia* [8], *C. limetta* [19], *C. medica* [22,23], *C. reticulata* [25], *C. tamurana* [28,29] |  |
| Nerol formate | *C. reticulata* [25] |  |
| Ketones | | |
| β-Ionone | *C. tamurana* [28] |  |
| Nootkatone | *C. bergamia* [14,15], *C. paradisi* [16,17] |  |
| Oxides | | |
| Caryophyllene oxide | *C. aurantium* [11,12], *C. bergamia* [15], *C. limon* [11], *C. maxima* [17], *C. paradisi* [17], *C. reticulata* [11,25], *C. sinensis* [11], *C. tamurana* [28,29] |  |

**Table 1.** *Cont.*

| Compound * | *Citrus* Species | Structure |
|---|---|---|
| Diterpenes | | |
| Geranyl α-terpinene | *C. sinensis* [27] | |
| *E*-Phytol | *C. reticulata* [25] | |
| Coumarins | | |
| Bergamottin | *C. bergamia* [13] | |
| Bergapten | *C. bergamia* [13] | |
| Citropten | *C. bergamia* [13] | |
| 5-Geranyloxy-7-methoxycoumarin | *C. bergamia* [13] | |
| Phenylpropanoids | | |
| Cinnamic aldehyde | *C. paradisi* [17] | |
| Cinnamyl alcohol | *C. tamurana* [28,29] | |
| α-Curcumin | *C. aurantifolia* [9] | |
| Estragole | *C. limon* [21] | |

**Table 1.** *Cont.*

| Compound * | *Citrus* Species | Structure |
|---|---|---|
| Ethyl cinnamate | *C. limon* [21] | |
| Ethyl p-methoxycinnamate | *C. limon* [21] | |
| Eugenol | *C. tamurana* [28] | |
| Isoeugenol | *C. tamurana* [28,29] | |
| Isosafrole | *C. reticulata* [26] | |
| Methyl eugenol | *C. maxima* [16] | |
| Miscellaneous | | |
| *E*-Solanone | *C. bergamia* [15] | |
| Sulcatone | *C. bergamia* [15], *C. limon* [21], *C. reticulata* [25], *C. tamurana* [28] | |

* Stereochemistry is reported when inferable from the original manuscripts.

## 3. Main Changes in the Essential Oils Content during Fruit Maturation

The fruit quality attributes (e.g., juice color, flavor, seed presence, shape, peel color, presence of alterations) have strong economical relevance because they are related to the consumer perception. It is known that the general nutritional properties vary remarkably during the fruit ripening stages and harvesting time as a result of variation in nature and concentration of organic acids, sugars and phenolics which largely affect taste and organoleptic quality [31]. Among these compounds, EOs have attracted attention due to their high content in peels, aroma, flavors and bioactive properties. Although different citrus species share many EOs (Table 1), each fruit has a distinctive odor that is related to the presence or absence of unique components, which significantly influence the organoleptic properties and, as a consequence, the market destination of the fruit [32].

Besides the genetic bases concerning species (Table 1) and cultivars [33], preharvest climatic conditions and cultural practices (e.g., rootstocks, irrigation, crop management following conventional or organic farming), harvesting time and methods represent crucial factors that affect the chemical composition of EOs. In this respect, it has been observed that the rootstock may influence the yield and composition of peel EOs in orange, with a low impact on flavor. Generally, neither the rootstock nor the scion ploidy levels affect the EOs content; however, the tetraploid level of the scion may significantly reduce the oxygenated compound fraction. Sensitive significant differences were detected between the reference sample (diploid scion–diploid rootstock) and the three other diploid-tetraploid combinations, suggesting that the rootstock and the ploidy level of the scion are key elements for the profiling of aromatic flavor [27].

In general, the fruit growth and development in citrus consists of three stages: a first phase of slow growth, a second phase of major increase in size and weight by growth of juice sacs from the pulp and a third phase of reduced fruit growth together with fruit transformation and maturation [34]. The decision on harvesting time is critical in defining the quality of the fruits.

In order to maximize the quality of citrus fruit in terms of EO production, studies have been conducted to investigate the evolution of the main products at the different stages of ripening [11,35–40], observing significant variation essentially in monoterpene hydrocarbons and oxygenated monoterpenes.

An investigation on variation during ripening of the chemical composition of the peel of four citrus (i.e., *C. aurantium*, *C. limon*, *C. sinensis*, *C. reticulata*) revealed that it depends on the stage and the species. In fact, the stage of maximum yield of EOs production is the immaturity for *C. limon*, the semimaturity for *C. sinensis* and *C. reticulata* and the maturity for *C. aurantium*. However, in all these species the highest level of limonene was already reached at the immature stage [11]. Subsequently, Bhuyan et al. [38] confirmed that in *C. reticulata* the maximum yield of peel oil is reached at the turning (semimature) stage, with lower specific gravity, refractive index and ester number, and a higher content in aldehydes and other organoleptically important oxygenated constituents; particularly, these products contribute to the overall quality and typical aroma of mandarin oil, making it more suitable for commercial applications.

An Italian study also demonstrated that variation of peel EOs during fruit ripening depends on the origin and cultivars. In fact, the measurement of EO content of four cultivars of *C. limon* at different harvesting times (i.e., October, November, December, February) showed that the maximum yield was reached in November for Campanian cultivars (i.e., 'Ovale di Sorrento' and 'Sfusato Amalfitano'), whereas in Sicilian cultivars (i.e., 'Femminello Cerza' and 'Femminello Adamo') the peak was reached in December. Furthermore, the most abundant monoterpene hydrocarbons (e.g., α-pinene, β-pinene, myrcene, D-limonene, and γ-terpinene) decreased during the ripening stages [36].

The EO yield was also determined for *C. medica* peels, observing a marked increase during fruit development and maturation. The content of some components, particularly limonene, α-thujene, 3-carene, α-pinene, β-pinene and γ-terpinene, varied significantly during maturation stages [35].

Investigating the effect of harvesting time on the volatile compounds produced by *C. bergamia* is particularly important considering that bergamot fruits are primarily used for the extraction of EOs employed in perfumes, cosmetics and confections. Marzocchi et al. [37] conducted a study aimed to assess the EO quality of two varieties of *C. bergamia* at different maturation stages, observing that the volatile compound concentration is higher at the second and the third stage. Particularly, the maturation stage seemed to affect many compounds (e.g., β-pinene, γ-terpinene, α-terpineol), while limonene, the most representative compound, had similar concentrations in all varieties regardless of the harvesting time.

## 4. Biological Properties as Related to Protection against Postharvest Deterioration

Many substances contribute to the biological properties of citrus fruits, such as carotenoids [41], flavonoids [42], ascorbic and other organic acids [31]. However, in consequence of the possibility of extracting them from the peels as by-products, bioactivities of EOs have been considered more in-depth, and independently studied. Besides usage in cosmetology [43], these properties have been essentially investigated with reference to multiple favorable effects on health and ensuing pharmaceutical relevance. This refers to possible applications in the treatment of neurological [44] and vascular disorders [45], and as antioxidant [12,46–50], anticholesterolemic [51], antidiabetic [12] and antitumor agents [43,52]. The latter effects have been also studied with reference to purified products, such as citral [53,54], bergamottin and 5-geranyloxy-7-methoxycoumarin [13].

However, the most exploited applications undoubtedly rely on antimicrobial properties (Table 2). These concern use of EOs not only as alternative antibiotics in the medical field [12,55–60], but particularly as food additives. The latter usage, that may also involve EOs extracted from other plants which are not considered in this article, aims at improving the quality of a wide array of food products with reference to both possible contamination with human pathogens and preservation of organoleptic properties after inhibition of deteriorating microbial agents [47,50,56,61–66]. Of course, the latter is a priority aspect in the case of citrus fruits, which generally need to overcome prolonged postharvest periods before undergoing either fresh consumption or industrial transformation. Hence, to a certain extent citrus fruits benefit from a natural protection against postharvest deterioration from the EOs which abound in their epicarp (Table 2).

**Table 2.** Bioactivity of essential oils extracted from citrus peels as related to fruit quality.

| Species | Bioactivity | References |
|---|---|---|
| *C. aurantifolia* | Acaricidal<br>Antibacterial<br>Antifungal<br>Insecticidal | [9]<br>[67,68]<br>[8,67,69–74]<br>[75] |
| *C. aurantium* | Antibacterial<br>Antifungal<br>Antioxidant<br>Insecticidal | [12,47,49,68,76,77]<br>[49,77,78]<br>[12,47,49]<br>[77,79] |
| *C. bergamia* | Antibacterial<br>Antifungal<br>Antioxidant<br>Insecticidal | [56,57]<br>[77,78,80]<br>[48]<br>[81] |
| *C. latifolia* | Antifungal | [8,70] |
| *C. limon* | Antibacterial<br>Antifungal<br>Antioxidant<br>Insecticidal | [47,49,50,55–57,61,62,64,66,67]<br>[8,49,50,55,62,63,67,69–71,73,78,82–84]<br>[46,47,49]<br>[33,79,81] |
| *C. limonia* | Antifungal | [8] |
| *C. maxima* | Antibacterial | [65] |
| | Antifungal | [70,74,85] |
| *C. medica* | Antibacterial | [86,87] |
| | Antifungal | [59,88] |
| *C. paradisi* | Antibacterial<br>Antifungal<br>Insecticidal | [64,67]<br>[63,64,67,69,70,73,89]<br>[33] |

**Table 2.** *Cont.*

| Species | Bioactivity | References |
|---|---|---|
| *C. reticulata* | Antibacterial | [47,49,50,60,64] |
| | Antifungal | [24,25,49,50,63,69,70,74,90] |
| | Antioxidant | [47,49] |
| | Insecticidal | [79] |
| *C. sinensis* | Antibacterial | [47,50,55–57,64,67,73,90,91] |
| | Antifungal | [50,55,63,67,70,73,74,85,88,90–97] |
| | Antioxidant | [47] |
| *C. unshiu* | Antibacterial | [30] |
| | Antifungal | [70] |

Postharvest quality of citrus fruits is affected by several fungal pathogens which start the infection process in the field, such as *Phytophthora citrophthora, Alternaria citri, Diaporthe citri* and *Lasiodiplodia theobromae* [98–100]. In the case of the tropical citrus spot agent *Pseudocercospora (=Phaeoramularia) angolensis*, tolerance by certain species/cultivars has been referred to the composition of peel EOs [70]. Several practical applications have dealt with the heterologous use of citrus EOs against biological adversities of other crops [89,92,101]; many examples concern pests and pathogens which are also known on citrus fruits, hence representing indirect evidence of possible effectiveness in their management on the latter. This is the case of reported effects against *Colletotrichum gloeosporioides*, the agent of citrus anthracnose also known as *Glomerella cingulata*, which damages fruits both in the field and in postharvest [71,72,88,96,102]. Inhibitory effects have also been documented against other fungi, such as *Aspergillus* spp. [55,63,78,82,85,93–95,97,101,103,104], which may affect citrus quality as mycotoxin producers [105,106]. The gray mold agent *Botrytis cinerea*, also recently affirmed as an emergent citrus postharvest pathogen [107], proved to be sensitive to lemon essential oil, which completely inhibits in vitro growth at concentrations of 17 μL mL$^{-1}$ [83], or 0.016% [108]. However, lower inhibitory activities resulted from two other studies carried out with EOs of *C. limon, C. limonia. C. aurantifolia* and *C. latifolia* (MIC 312–625 μg mL$^{-1}$) [8], and of *C. aurantifolia, C. limon, C. paradisi* and *C. sinensis* (EC$_{50}$ ranging between 249 and 809 mg L$^{-1}$) [67], with the latter also showing mild antibacterial effects. Inhibitory properties against the agent of bacterial canker *Xanthomonas citri* subsp. *citri* were displayed by EOs extracted from *C. aurantium* [76], from *C. aurantifolia* and *C. aurantium* [68] and from *C. reticulata, C. limon, C. sinensis* and *C. aurantium* [47].

Undisputedly, a major impact on citrus fruit quality has to be ascribed to the agents of blue mold (*Penicillium italicum*) and green mold (*Penicillium digitatum*) [109]. EOs from orange (cvv. 'Washington Navel', 'Sanguinello', 'Tarocco', 'Moro', 'Valencia late' and 'Ovale'), bitter orange, mandarin (cv. 'Avana'), grapefruit (cvv. 'Marsh seedless' and 'Red Blush') and lemon (cv. 'Femminello', collected in three periods) displayed to various extents inhibitory activities against these fungi, with *P. digitatum* being more sensitive [69]. Moreover, treatments with lime EOs at 10% concentration reduced disease severity on orange fruit, along with significant reduction of the undesirable fruit percentage and weight loss, and increase in total soluble solid content during cold storage for 14 weeks [73]. Young green lemon fruit show a significantly lower level of postharvest decay as compared to the older yellow fruit. Inoculation with *P. digitatum* demonstrated that resistance of green fruit is related to compounds synthesized in the oil glands of the flavedo, the majority of which are identified as the monoterpene aldehyde citral. Flavedo of green lemons contains 1.5–2-times higher levels of citral than the yellow fruit. In parallel with citral decline, flavedo extracts of yellow lemons exhibited an increased level of the monoterpene ester neryl acetate, which not only proved to be inactive but, in concentrations below 500 ppm, even stimulated development of *P. digitatum*. During long-term storage, citral concentration decreases in parallel with the decline of antifungal activity in the peel and with an increase of decay incidence [110]. Citral was recently found to affect ergosterol

biosynthesis, indicating general antifungal effects [111]. In fact, along with its isomers geranial and neral, it also displayed various amounts of inhibitory effects against *P. italicum* and *Geotrichum candidum* [112–114], as well as α-terpineol [115].

Other purified EOs have shown inhibitory effects against *P. digitatum*. Citronellal inhibited mycelial growth and spore germination in a dose-dependent manner, with a MIC of 1.60 μL mL$^{-1}$ and minimum fungicidal concentration (MFC) of 3.20 μL mL$^{-1}$ [116]. When assayed as encapsulated oil-in water nanoemulsions, eugenol, carvacrol and cinnamaldehyde displayed antifungal effects in a dose-dependent manner with MIC of 0.125 mg mL$^{-1}$ and MFC of 0.25 mg mL$^{-1}$. Nanoemulsion coating reduced fruit decay, weight loss and respiratory rate; degradation of soluble solids, vitamin C and titratable acids were delayed, while antioxidant enzyme activities were significantly increased and maintained during postharvest storage [117]. Moreover, *R*-(L)-carvone completely inhibited mycelial growth at a concentration of 1000 μL L$^{-1}$ and caused approximately 60% inhibition at 500 μL L$^{-1}$, while 1,8-cineole was effective at 3000 μL L$^{-1}$, but inhibition decreased to 83% at 2000 μL L$^{-1}$; (D)-limonene was ineffective against this pathogen with only 50% inhibition achieved at 3000 μL L$^{-1}$ [118]. Otherwise, limonene has been reported for general fungistatic properties as assayed on a panel of five test fungi; however, lower fungitoxicity than the crude extract is indicative of synergistic action by other EO components [92,104]. The compound also showed direct anti-aflatoxigenic properties [85]. Thymol and carvacrol exhibited strong antifungal activity against *B. cinerea*, with MIC and MFC of 65 mg L$^{-1}$ and 100 mg L$^{-1}$ for thymol, and 120 μL L$^{-1}$ and 140 μL L$^{-1}$ for carvacrol [119]. In another study (D)-β-pinene, (L)-α-pinene, (L)-β-pinene, (D)-limonene and (L)-limonene displayed some inhibitory effects against the gray mold agent [8]. Citral presented MIC of 0.5% on *A. niger*, *A. flavus* and *Fusarium* sp., 2.0% on *Penicillium* sp. and 8.0% on *Rhizopus* sp., while eugenol presented MIC of 2.0% on *A. flavus* and 4.0% on *Penicillium* sp., *Fusarium* sp. and *Rhizopus* sp. [82]. Some of these products, such as *D*-limonene, have also displayed insecticidal and acaricidal properties [9,120], which can result in a protective effect against insect pests both in the field and in postharvest handling.

Some typical EO constituents, such as thymol, eugenol and geraniol, have been recently assayed as nanoemulsions, displaying bactericidal or bacteriostatic effects against the citrus pathogens *Xanthomonas fuscans* subsp. *aurantifolii* and *X. citri* subsp. *citri* [121]. Inhibitory effects against the latter were also documented for α-terpineol, citral, citronellal, geraniol, linalyl acetate and linalool [68]. Particularly, the latter compound has been shown to mediate resistance against bacterial canker in mandarin [122] and in transgenic sweet orange [123].

As a counterpoint to the above favorable effects, another study pointed out that EOs of *C. limon, C. limonia, C. aurantifolia* and *C. latifolia* enhance *P. digitatum* in vitro, and that growth of the fungus was even stimulated by pure chiral volatile compounds from these EOs, apart from citral and (+)-β-pinene [8]. Likewise, EOs of mandarin stimulated spore germination and mycelium growth of *P. digitatum* and *P. italicum* at a low concentration, but they were strongly inhibitory at a higher concentration [124]. This hormetic effect has been also displayed by selected compounds of EOs, such as citral and linalool [125]. Indeed, some evidence has been gathered that compounds in the EOs may act as odor cues for attracting pest or inducing citrus plant pathogens. In fact, both *P. digitatum* and *X. citri* subsp. *citri* were unable to infect the peel tissues of transgenic orange fruits with downregulation of limonene synthase and reduced accumulation of limonene in the peel [126].

Contrasting results have also been obtained with reference to effects against the Mediterranean fruit fly (*Ceratitis capitata*). In fact, EOs are generally considered as the most critical resistance factor to medfly infestation of various citrus fruits. Toxic effects on larvae were reported to be caused by treatments with EOs extracted from some varieties of lemon, sweet and bitter orange. The two latter species, presenting a higher limonene concentration, produced stronger effects, while presence of α-pinene and β-pinene was considered to account for the lower toxicity of lemon EOs [81]. Likewise, ether extracts from lemon and grapefruit peel proved to be toxic to the eggs and larvae of both the medfly



and the South American fruit fly (*Anastrepha fraterculus*), along with purified limonene and citral. There were no or less significant differences in toxicity of extracts from ripe and overripe fruit of both species [33]. As the ovipositional responses of female medflies were investigated in dual-choice experiments, a significantly higher number of eggs was laid into hollow oviposition hemispheres which had been pre-punctured with 1 µL of peel oil from sweet orange, satsuma mandarin, bitter orange, grapefruit and lemon. The latter had just a weak stimulatory effect, while sweet orange oil was the most active in eliciting oviposition. Limonene stimulated oviposition, whereas linalool, a representative compound of immature citrus fruit associated with high toxicity against immature stages of fruit flies, had a significant deterrent effect. In further no-choice tests, females laid about 23% fewer eggs in limonene 93% (that is the amount found in orange oil) and 60% fewer eggs in limonene 93% plus linalool 3% (approximately 10-fold the amount found in orange oil) mixtures, as compared to sweet orange oil. Hence, the limonene content accounts, largely but not completely, for the ovipositional responses observed in sweet orange oil, whereas high linalool proportions are capable of significantly masking and/or disrupting its stimulatory effects [127]. Limonene, linalool and α-pinene induced toxicity to adult medflies, with males being more sensitive than females. Sub-lethal doses of limonene ($LD_{20}$) enhanced the lifespan of medflies when they were deprived of protein and positively affected fecundity, implying positive effects of sub-lethal doses at least at certain stages; hence, a general hormetic-like response [128]. Detrimental effects on oviposition were observed on linalool-treated bitter oranges. Particularly, oranges that were offered to females immediately after exposure to linalool received more oviposition stings and eggs than those offered three days post-exposure. More flies were captured in traps placed on untreated-control than on linalool-treated trees. Spraying and topical-droplet application were more efficient than exposure to vapors of linalool in ethanolic solution [129]. Reduction of oviposition was also observed on oranges treated with lemon EOs, with some differences between oils extracted from the the two cultivars 'Interdonato' and 'Lunario', the first one being more effective at lower dose [130]. Significant improvement was also observed on the mating behavior of medfly males, after exposure to commercial EOs from peels of bitter orange, mandarin orange, lemon and grapefruit; more particularly by doses of 12.5 or 25 µL of sweet orange oil. Moreover, a mixture of geraniol, α-pinene, limonene, β-myrcene and linalool also determined mating advantages. Considering that this mixture did not contain α-copaene, an EO known to enhance mating success, it could find alternative application in view of a more cost-effective and efficient implementation of the sterile insect technique in the integrated management of *C. capitata* [131]. Among the above compounds, further studies have shown that linalool is particularly relevant in this attractive effect [132].

Matching what was previously reported for citrus pathogens, the general remark that medflies are less attracted by low limonene-expressing fruit indicates that accumulation of this main component of EOs in the peel of citrus fruit is involved in the successful trophic interaction between fruit, insects and microorganisms. Hence, terpene downregulation has been proposed as a strategy to generate broad-spectrum resistance against the pests and pathogens of these crops. However, this issue requires careful thought, considering that limonene is recognized to play a role in citrus defense against other pests, such as scales, whiteflies and mealybugs [133].

The above-mentioned antioxidant properties of EOs, documented in the case of extracts from several citrus species [12,46–50], should also be considered with reference to their preservative action on fruit in postharvest. EOs and other volatile organic compounds (VOCs) have been proposed and routinely used as biofumigants for the treatment of several food commodities, including fruit where postharvest deterioration caused by molds is a general problem affecting trade and storage [134]. Citrus peel extracts are generally recognized as safe products to be used for treatment of commodities, including wheat seeds, where they do not cause adverse effects on germination [93], and in the fresh market of fruit and vegetables as bioactive edible coatings [135]; the latter also in combination with other effective products such as chitosan [98,136–138].

## 5. The Role of Endophytic Fungi

Endophytes are recognized as a key factor impacting plant health. Moreover, by influencing physiology of reproductive organs, they govern the process of fruit ripening, with ensuing effects on quality and preservation during marketing [139]. On the other hand, microbiome components functionally interact with each other in multiple ways, in connection with factors such as the plant genotype, the developmental stages and the environmental conditions [140].

It is generally accepted that many endophytic fungi effectively improve citrus plant fitness by promoting growth and participating in defensive mutualism [141,142]. At least in part, the latter function is due to the capacity of the endophytic strains to release bioactive metabolites. This property is not only referable to peculiar antimicrobial products, but also to several compounds which are originally known as plant secondary metabolites, including some belonging to the EOs [143]. Indeed, the long list of endophytic fungi reported in this reference published in 2015 as able to directly synthesize these compounds should be integrated with an even higher number of findings resulting from the manifold studies carried out throughout the world in the last seven years, attesting the increasing awareness of the importance of this component of biodiversity. Moreover, endophytes have been found to be involved in biotransformation of products in EO fraction, as experimentally demonstrated in the case of *Piper aduncum* [144]. On the other hand, biosynthesis of EOs by some endophytes may be induced in the presence of plant pathogens. This is the case of a strain of *Trichoderma longibrachiatum* producing cedrene, epi-β-caryophyllene and nerolidol in co-culture with *Fusarium oxysporum* [145].

With specific reference to *Citrus* spp., the ability of strains of *Annulohypoxylon* sp. from *C. aurantifolia* to produce 1,8-cineole [146] and of *Muscodor* sp. from *C. sinensis* to synthesize *cis*-, *trans*-α-bergamotene, cedrene, Z-β-farnesene and other volatile compounds which collectively inhibit the agent of citrus black spot (*Phyllosticta citricarpa*) [147], confirms the inference that endophytic fungi may effectively support and integrate the pool of products synthesized by the host plant. The relevance of these findings is corroborated by a general biosynthetic aptitude of xylariaceous fungi, which are widespread as endophytes and, besides *Muscodor* (syn. *Induratia*) and *Annulohypoxylon*, include the genera *Xylaria*, *Daldinia* and *Hypoxylon* (syn. *Nodulisporium*), also reported as EO producers [141].

A practical application derived from basic observations concerning these fungi has been named mycofumigation [148]. It consists of the treatment of food commodities with the volatile fractions of extracts from cultures of fungal strains containing antimicrobial products which act by permeating the surrounding atmosphere during product storage. The importance of this technique in postharvest handling of citrus fruits is intuitive, and one practical experience reported for the endophytic isolate CMU-UPE34 of *Nodulisporium* sp. is valuable for elucidating the expected impact on the product quality. In fact, this strain was found to effectively inhibit growth of two citrus molds (*P. digitatum* and *Penicillium expansum*) by releasing a mixture of 31 VOCs containing eucalyptol (1,8-cineole) and terpinen-4-ol as the most abundant products [149]; these compounds act synergistically, as it has been demonstrated against *B. cinerea* [150].

Fumigation of oranges with citral (20, 60 or 150 mL L$^{-1}$ in absorbent pads) in a closed system, following application of conidia by puncturing, delayed the onset of sour rot by *G. candidum* var. *citri-aurantii* at room temperature by 7–10 days and at 5 °C, by 13–30 days, but had limited effect on blue and green mold, which developed faster on oranges wounded by puncture than by abrasion. Volatile citral delayed the development of blue mold in abraded, but not punctured, oranges stored at 5 °C. Phytotoxicity symptoms were observed on the upper surface of some fruit close to or in direct contact with citral-soaked pads at concentrations of 60 and 150 mL L$^{-1}$. Citral residue was not detected in the rind of fumigated oranges. Volatile citral applied at 60 mL L$^{-1}$ appeared to have potential for the control of sour rot, although phytotoxicity was associated with high concentrations [151].

So far, limited data are available on the biosynthetic capacities of citrus endophytes. Hence, supplementary work must be carried out to verify properties by the most common

endophytic species and to assess if they can play a significant role in driving the ripening process and preserving fruit quality both in the grove and in postharvest. Indeed, the direct use of endophytic fungi may represent a more effective tool than the classic treatments with selected chemicals, considering that these microbial associates are capable of continuously releasing a bulk of substances acting in synergism, and that the concomitant presence of multiple bioactive compounds may minimize the risk of induction of resistance in the challenged pathogens.

## 6. Conclusions

In recent years, citrus essential oils have gained great popularity for applicative usage in the food and cosmetic as well as the pharmaceutical industries, which has been technically refined by the availability of several kinds of microformulations [152]. The increasing demand underlines the opportunity to recycle citrus peels contained in wastes from the food industry for their extraction [80,153,154]. However, the raw material is qualitatively highly heterogeneous as it is influenced by several factors including the nature and provenance of the fruit, genotype, soil type, climatic and cultural conditions. It has been demonstrated that the chemical compositions of the citrus peels vary significantly during ripening. For this reason, harvesting time is a critical parameter, especially because EOs have an important role in the protection from postharvest fruit deteriorations. In fact, EOs in the immature fruit stages have a higher effectiveness against pests and pathogens. This evidence may be adaptatively interpreted with the necessity by the plant to protect developing fruits; while in ripe fruits this need would no longer be relevant, considering that they are destined either to be eaten by frugivorous animals or to fall to the ground to allow seed dispersal [155]. Adaptative factors may also regard the capacity of some endophytic associates to directly synthesize these products, which contribute to shaping their ecological role as defensive mutualists. Undoubtedly, this aptitude deserves more in-depth assessments in view of a possible exploitation for improving postharvest quality.

**Author Contributions:** Conceptualization, R.N. and A.A.; writing—original draft preparation, M.M.S.; writing—review and editing, M.M.S., R.N. and A.A. All authors have read and agreed to the published version of the manuscript.

**Funding:** This research received no external funding.

**Institutional Review Board Statement:** Not applicable.

**Informed Consent Statement:** Not applicable.

**Data Availability Statement:** Not applicable.

**Conflicts of Interest:** The authors declare no conflict of interest.

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
