# Peer review of "Essential Oils in Citrus Fruit Ripening and Postharvest Quality"

_horticulturae, doi:10.3390/horticulturae8050396_

Round 1

Reviewer 1 Report

The paper entitled “ Essential Oils in Citrus Fruit Ripening and Postharvest Quality by Maria Michela Salvatore, Rosario Nicoletti  and Anna Andolfi Joanna combines and provides an overview of the importance of individual species of the genus Citrus, ie the composition of essential oils depending on the phase of growth and ripening, ie fruit and quality after harvest.
The topic is very current today, and the basic idea of the manuscript is good and could be of practical interest given the possibility of better use of the peel of Citrus.
In my opinion, this paper can be accepted after minor revision for publication in a Journal - Horticulturae.  

Detailed comments are given in the manuscript.

Author Response

The paper entitled “ Essential Oils in Citrus Fruit Ripening and Postharvest Quality by Maria Michela Salvatore, Rosario Nicoletti  and Anna Andolfi Joanna combines and provides an overview of the importance of individual species of the genus Citrus, ie the composition of essential oils depending on the phase of growth and ripening, ie fruit and quality after harvest.
The topic is very current today, and the basic idea of the manuscript is good and could be of practical interest given the possibility of better use of the peel of Citrus.
In my opinion, this paper can be accepted after minor revision for publication in a Journal - Horticulturae.  

Detailed comments are given in the manuscript.

Thank you for your positive comments.

Line 26 - I suggest that the structure of the citrus fruit be described in more detail - which is albedo, flavedo, oil glands - the size, shape of the oil glands and the content of the essential oil depend on the type and cultivar of citrus

Concerning fruit structure, we preferred not to get into details considering that our paper deals with all Citrus spp. and that, as pointed out in your comment, it may vary among species and cultivars.

Line 144 – The name of the cultivars should be written in single quote - e.g. ‘Ovale di Sorrento’- correct so on through the text

Done.

Line 378 - to single out which type of citrus has the greatest potential for a particular purpose with regard to the composition, properties and application of EOs

Considering the varied composition of EOs and the several factors affecting their production, there is not a citrus species with the greatest potential. Moreover, our review does not specifically consider biotechnological applications.

Reviewer 2 Report

Dear authors,

I read your manuscript entitled "Essential Oils in Citrus Fruit Ripening and Postharvest Quality". Unfortunately, i believe that it can not be published and has to be rejected.

The manuscript faces "structural" issues. At first, only a very small part of the manuscript is related to ripening and i could not find any factor affecting (as you mention in abstract) except from ploidy number which is discussed in the segment 3 but shortly. Moreover, there are large parts with irrelevant information (e.g. the segment referred to Ceratitis capitata), unnecessary information (such as the mls added for Botrytis cinerea in in vitro culture) and others.

It some cases the manuscript makes me the sense it is out of the topic. In addition, more attention is paid on the fungi and how they are affected by EOs than the role of EO on postharvest quality of the fruit. Focusing only on EOs and fungi could be a separate review article but more information should be added. Probably, changing the title and re-writting accordingly would be the best choice.     

In general, stay in topic straight to the point and be consistent. It is better to have a smaller but more precise manuscript. In any case, i have to reject the manuscript.    

Author Response

Thank you for your comments. However, we consider that they are not in line with those from other three referees.

Reviewer 3 Report

The article was very well/clearly designed and written correctly. It presents the latest knowledge on the composition of diverse citrus essential oils and on the factors influencing their composition. I recomand to publish the work in the current form.

Author Response

The article was very well/clearly designed and written correctly. It presents the latest knowledge on the composition of diverse citrus essential oils and on the factors influencing their composition. I recomand to publish the work in the current form.

We wish to thank the referee for the positive comments.

Reviewer 4 Report

An interesting issue is described in the review related to the active compounds contained in citrus fruits.

The title and summary should be expanded because it does not contain the whole essence of the review. In the work, there is a large disproportion in a very detailed description of some research results, and the reference to other issues on a general basis (indicated in the text).
Some irreconcilable dependencies should be commented on, eg the content of compounds and the quality and yield of citrus fruits. The active compounds are extracted on the basis of a waste product (fruit coating).

Author Response

An interesting issue is described in the review related to the active compounds contained in citrus fruits.

Thank you for your positive comment.

The title and summary should be expanded because it does not contain the whole essence of the review. In the work, there is a large disproportion in a very detailed description of some research results, and the reference to other issues on a general basis (indicated in the text).
Some irreconcilable dependencies should be commented on, eg the content of compounds and the quality and yield of citrus fruits. The active compounds are extracted on the basis of a waste product (fruit coating)

Title – Add content… and properties against fungi and pest

Discussing properties of EOs as related to postharvest quality incorporates aspects concerning effects against deteriorating agents; hence, we think that the title is exhaustive.

Summary - The summary does not reflect the entire literature review.  And where the use of active compounds in protection against diseases and pests, which is an important part of the literature review

The summary was modified as suggested by the referee.

Line 84 - how the other factors mentioned affect the active compound content, the effect of the rootstock is only briefly discussed

We understand the sense of this comment; however, a review paper is based on the existing literature, and it is quite obvious that different subjects/aspects cannot be treated with a similar level of details.

Line 107 - how to reconcile the content of compounds with the ripeness of fruit for consumption, this issue requires a comment

Modified as suggested.

Line 117 – comment as above

Modified as suggested.

Line 224 - I am asking for specific applications, in which plant species these compounds were used, what effect was obtained, everything is at the level of generalizations

As specified in the text, in this review we only examined data concerning the use of citrus oils on citrus plants.

Line 269 - A very detailed description of some research results, and others are only mentioned, why this disproportion

Answer as Line 84.

Line 305 - please give examples of seeds of what plant species, what species of fruit and vegetables

We specified that observations on absence of effects on germination refer to wheat seeds, while the reference to use in coatings on fruit and vegetables is general.

Conclusion – The conclusion is too general and repetitive, it is not necessary to quote the authors, but to extract a few key sentences.

Line 382 - whether it is compatible with the ripeness of the fruit for harvest

No correlation has been pointed out concerning this aspect.